

# Knowledge, attitude and purchasing behavior of Saudi mothers towards food additives and dietary pattern of preschool children

Reem H. Almoabadi and Mahitab A. Hanbazaza

Department of Food and Nutrition, Faculty of Human Sciences and Design, King Abdulaziz University, Jeddah, Saudi Arabia

## ABSTRACT

**Background.** There are over 506 children's products containing one or more types of additives. Maternal awareness of these additives is essential for the health of preschool-aged children, as this period is vital for children's growth and development. This study aims to assess the knowledge, attitudes, and purchasing behaviors related to food additives among mothers living in the western region of Saudi Arabia, as well as the dietary patterns of preschool children.

**Method.** A cross-sectional study was conducted using an online survey with a convenience sample of 521 mothers of preschool-aged children (3–5 years old). The survey gathered data on the child's age, number of children, the youngest child's weight and height, food intolerance, tooth decay, as well as the dietary patterns of preschool children. It also assessed the mother's knowledge, attitude, and purchasing behaviors related to food additives.

**Results.** The study found that 46.6% of mothers demonstrated good knowledge of food additives, while 56.0% demonstrated fair attitudes and 78.5% good purchasing behavior regarding additives. Additionally, the majority of mothers reported favorable dietary patterns for their preschool-aged children. "Biscuits and crackers" had the highest consumption frequency ($4.98 \pm 1.50$), with 36.7% of children consuming them once daily, while "Soft beverages" had the lowest consumption frequency ($2.73 \pm 2.04$), with 46.6% of children never consuming them. Statistically significant differences were identified between mothers' knowledge and their age, education level, occupation status, and economic status ($p < 0.05$). ANOVA results also indicated a statistically significant difference between mothers' attitudes and occupation status ($p < 0.05$). Furthermore, there were significant positive correlations between mothers' knowledge of food additives and their attitudes ($r = 0.293$) and purchasing behaviors ($r = 0.284$) related to additives.

**Conclusion.** The findings suggest that mothers possess a relatively good level of knowledge of food additives and hold fair attitudes toward them, tending to result in healthier purchasing behaviors and dietary practices for their preschool-aged children. To increase awareness, nutrition intervention programs are required across various socio-economic groups of mothers in the western region of Saudi Arabia. These programs can significantly contribute to promoting healthier dietary practices for preschool-aged children and improving overall family health and well-being.

Corresponding author
Mahitab A. Hanbazaza,
mhanbazaza@kau.edu.sa

## INTRODUCTION

In recent years, food additives have gained increasing importance in the food industry. Food additives are defined as substances added to food to enhance or alter flavor, texture, and appearance. Initially, the primary purpose of food additives was to extend shelf life, enhance color, and add sweetness to products (*Kang, Kim & Kim, 2021*). Historically, natural additives such as vinegar, salt, and lemon were commonly used. However, with advancements in the food industry, artificial additives have largely replaced natural ones.

The standards, regulations, and usage of food additives are globally governed by the Codex Alimentarius, a program of the United Nations Food and Agriculture Organization (FAO) and the World Health Organization (WHO). The Joint FAO/WHO Expert Committee on Food Additives (JECFA), comprising experts from various countries, determines regulations and standards related to food additives (*Kraemer et al., 2022*). There are over 10,000 allowed chemical substances classified as food additives, including sodium nitrite (E 401), monosodium glutamate (E 621), high fructose corn syrup (HFCS), guar gum (E 412), trans fats, and artificial sweeteners (*Chazelas et al., 2020*). Globally, the presence of "E numbers" or symbols on labels of food products indicates the inclusion of additives. According to standards for labeling food additives set by the Saudi Food and Drug Authority (SFDA), product labels must include all details regarding the name, net contents, number of food additives (E number), side effects, and alcohol-free status (*Hussein, 2020*).

In supermarkets, over 506 products are categorized as children's, and this number is expected to increase in the future. Additionally, each food product may contain one or more types of additives (*Lorenzoni, Oliveira & Cladera-Olivera, 2012*). Childhood is a critical period for growth and development, and the high consumption of products such as candy, soft drinks, breakfast cereals, and jellies among preschool children can adversely affect their health and growth, leading to conditions such as diabetes and obesity (*Leal et al., 2017*; *Gulliford, 2008*). Additionally, some studies have shown a negative effect of food additives on behavioral changes in preschool children, such as hyperactivity (*McCann et al., 2007*). High intake of products with food additives has also been linked to dental caries (*Av, Adeloye & Ma, 2015*).

Mothers play an important role in shaping their children's food choices and dietary habits, particularly during the preschool years. According to *Kostecka (2014)*, preschool children consume more processed foods containing food additives than adults. A randomized, double-blind, placebo-controlled study analyzed the association between the intake of artificial food colorants and additives and behavioral issues in children, including hyperactivity and attention deficit disorders. The study concluded that certain food colorants might increase these conditions in susceptible children (*McCann et al., 2007*). Several reviews have also indicated an association between various food additives and

long-term health effects in children, such as endocrine disruption, developmental delays, increased susceptibility to chronic diseases, hyperactivity, allergic reactions, and metabolic disturbances, raising particular concern over the cumulative effects of prolonged exposure to these additives on overall health (*Savin et al., 2022*; *Trasande, Shaffer & Sathyanarayana, 2018*). Therefore, proper knowledge, attitudes, and purchasing behaviors regarding food additives among mothers are critical to the health of children. When mothers are educated about healthy food choices, their children are more likely to consume healthier food options compared to children of mothers who lack this awareness. A systematic review of the effects of culinary interventions on dietary intake and behavioral change indicated that culinary interventions were associated with improved attitudes, self-efficacy, and dietary intake in adults and children (*Hasan et al., 2019*).

There is a noticeable gap in the literature specifically focusing on the knowledge, attitudes, and purchasing behaviors regarding food additives among mothers in the western region of Saudi Arabia. Therefore, this study aims to evaluate the knowledge, attitude, and purchasing behaviors related to food additives among mothers in this region, as well as the dietary patterns of preschool children. The results of this study are expected to assist policymakers in designing suitable, evidence-based policies regarding food additives and establishing appropriate health education programs for mothers.

## MATERIAL AND METHODS

### Design and study population

This study employed a quantitative descriptive cross-sectional design, utilizing a non-probability convenience sample of mothers with preschool children (aged 3–5 years) living in the western region of Saudi Arabia. The western region of Saudi Arabia is characterized by its diverse population, which may influence food consumption patterns and attitudes toward food additives, thus providing a broad perspective on individual behaviors. The sample size consisted of 385 respondents, with a margin of error of 5% and a confidence level of 95%.

### Procedure

Data were collected using an online survey, which was accessible from April 9, 2022, to January 1, 2023. The survey was disseminated by the Deanship of Graduate Studies and Scientific Research to the emails of students and faculties. Additionally, the survey was distributed in 56 kindergarten centers across various cities in the western region of Saudi Arabia. To reach a wider audience within the region, targeting different segments of the population, the survey was also distributed through social communication platforms such as WhatsApp, Telegram, and Twitter.

Electronic informed consent was obtained from participants prior to them filling out the survey. The survey was reviewed to ensure the questions were suitable for evaluating knowledge, attitudes, purchasing behaviors, and dietary patterns related to food additives. Face validity testing was conducted with eight mothers, and content assessment was conducted by six nutrition experts to ensure clarity and relevance of the survey.

The research protocol was approved by the Research Ethics Committee, Faculty of Medicine, King Abdulaziz university with reference number (184-22).

## Measurement

- **Demographic characteristics:** The survey included 12 questions covering demographic information related to the mother's age, education level, marital status, occupation, economic status, city of residence, child's age, number of children, and self-reported weight and height.

- **Mothers' knowledge of food additives:** This section included 10 questions assessing mothers' knowledge of food additives, developed based on a review of previously published questionnaires in the scientific literature (*Ibrahim et al., 2021*; *Moselhy et al., 2016*; *Slavica & Radoslav, 2013*). The total score ranged from 0 to 8 points, categorized as good knowledge (8–6 points), fair knowledge (5–3 points), and poor knowledge (2–0 points).

- **Mothers' attitudes toward food additives:** Eight questions were included to evaluate mothers' attitudes toward food additives in food products, developed based on a review of previously published questionnaires in the scientific literature (*Esfahani, Ziaei & Esfandiari, 2021*; *Zugravu et al., 2017*). The scoring ranged from 0 to 8 points, with categories defined as good attitude (8–6 points), fair attitude (5–3 points), and poor attitude (2–0 points).

- **Mothers' purchasing behavior:** Mothers' purchasing behavior was assessed using eight questions, with scoring ranging from 0 to 8 points and categories defined as good purchasing behavior (8–6 points), fair purchasing behavior (5–3 points), or poor purchasing behavior (2–0 points). These questions were developed based on a review of previously published questionnaires in the scientific literature (*Ali et al., 2019*; *Esfahani, Ziaei & Esfandiari, 2021*; *Slavica & Radoslav, 2013*; *Suparmi, Desanti & Cahyono, 2015*).

- **Dietary Patterns of Preschool Children:** Preschool children's eating behaviors were assessed using a food frequency questionnaire (FFQ), adopted from the study conducted by *Hamed, Ferer & Nofal (2021)*. The FFQ measured the frequency of consumption of various food products containing food additives across different categories, such as biscuits and crackers, chips, cake, cereals, flavored dairy products, chewing gums, canned juices, soft drinks, candy, chocolate, ice cream, jelly, processed meat, sauces, and instant noodles and soup. The frequency of consumption of each food was assessed across seven categories: more than once per day, once per day, 4–5 times per week, 2–3 times per week, once per week, once or less per month, and never.

## Data analysis

All data were analyzed using the Statistical Package for the Social Sciences (SPSS) for Windows, Version 29. Descriptive analysis was employed to provide information on basic characteristics of the data, including descriptive statistics, measures of central tendency, and dispersion. Normality tests were conducted on all study variables, including mothers' knowledge of food additives, attitudes toward food additives, purchasing behaviors, and preschool children's dietary patterns. A one-way ANOVA test was performed to compare

the total mean scores of mothers' knowledge and attitudes toward food additives according to demographic variables. *Post-hoc* multiple comparisons (Scheffé's method) tests were used for pairwise comparisons among age groups. Furthermore, the correlational method was used to determine the relationship between the study variables. The level of statistical significance was set at $p < 0.05$.

## RESULTS

### Socio-demographic data

The initial total sample size was 521 participants. However, 51 responses were excluded from the analysis due to participants residing outside the western region of Saudi Arabia, such as in Riyadh, the eastern region, the northern region, or the southern region. Among the remaining participants, the majority (94%) were married, 66.8% had at least one preschool-aged child (3–5 years old), and 63.3% lived in the city of Jeddah. Additionally, 56.6% of the mothers held a bachelor's degree, and 55.3% were housewives. In terms of children's ages, 42.6% of the mothers had a child aged 5 years, and 38.8% reported a monthly income ranging from 8,000 to less than 15,000 Saudi Riyals (SR). The highest proportion of participants, 25%, fell within the age range of 28 to 31 years, while the lowest proportion, 7.5%, was in the age range of 18 to 22 years. Regarding children's health indicators, 64.2% of the children were underweight, 14.2% were obese, 16.2% suffered from food allergies, and 28% had dental caries. There were 15 participants missing data for weight and 94 missing data for height in the questionnaire responses.

### Mothers' knowledge of food additives

Regarding the mothers' knowledge of food additives, 60.1% of participants had heard about food additives. Among them, 52.1% reported having acquired information about food additives from social media or the internet. Additionally, 68.1% of participants knew that additives are substances added to processed food to improve flavor, color, texture, and extend preservation. Similarly, 68.1% were aware that food additives include preservatives, coloring agents, antioxidants, and emulsifiers. A majority of participants (54.1%) believed that food factories adhere to specific global regulations regarding the authorized proportions of food additives allowed in their products. Moreover, 66.6% knew that food factories in Saudi Arabia have a clear list of food additives in their food products. Nearly half of the participants (47.85%) were aware that the type of food additive is written on the packaging and symbolized by the letter "E". Similarly, 41.5% knew that the numbers following the letter "E" denote the type of food additive contained in the product (*e.g.*, "E129"). Additionally, 66.8% knew that food additives in processed food products are categorized into natural and synthetic additives. Lastly, 72.4% were aware that excessive consumption of industrial food additives could lead to hyperactivity in children and other serious illnesses. Overall, most mothers had either fair (32.6%, 4.21 $\pm$ 0.768) or good knowledge (46.6%, 7.07 $\pm$ 0.836) of food additives.

Table 1 presents the total score for participants' responses to the eight questions on mothers' knowledge of food additives.

**Table 1  Total score of mothers' knowledge toward food additives.**

|       |      | *n* | % | Minimum | Maximum | Mean | Std. Deviation |
|-------|------|-----|----|---------|---------|------|----------------|
|       | Poor | 108 | 20.7 | 0 | 2 | .90 | .853 |
| Level | Fair | 170 | 32.6 | 3 | 5 | 4.21 | .768 |
|       | Good | 243 | 46.6 | 6 | 8 | 7.07 | .836 |
|       | Total | 521 | 100.0 | 0 | 8 | 4.85 | 2.518 |

Notes.
  The answer "Yes" was the correct answer and given 1, while "No" or "I don't know" were considered as an incorrect answer and given 0. The total score ranged from 0 to 8 points. The classification of scoring is dependent on participants response as: the scored from 8 to 6 points had good knowledge, from 5 to 3 had fair knowledge, and from 2 to 0 had poor knowledge.

**Table 2  Total score of mothers' attitudes toward food additives.**

|       |      | *n* | % | Minimum | Maximum | Mean | Std. Deviation |
|-------|------|-----|----|---------|---------|------|----------------|
|       | Poor | 105 | 20.2 | 0 | 2 | 1.47 | .651 |
| Level | Fair | 292 | 56.0 | 3 | 5 | 3.99 | .802 |
|       | Good | 124 | 23.8 | 6 | 8 | 6.35 | .544 |
|       | Total | 521 | 100.0 | 0 | 8 | 4.04 | 1.769 |

Notes.
  The scoring of this section was 1 point was given for each correct response and 0 points were given to each incorrect answer. The scoring in the attitude section ranged from 0 to 8. The participants who have scoring points from 8 to 6 was reflected as good attitude, the scoring points from 5 to 3 had fair attitude and the score points from 2 to 0 had poor attitude.

## Mothers' attitudes toward food additives

Regarding attitudes, 57% of mothers reported not allowing their children to consume foods containing food additives. Additionally, 71.2% stated that they lacked sufficient nutritional information to select suitable food products, including those with food additives. Furthermore, 65.1% believed that foods containing artificial additives are more attractive to consumers, while 66.4% were unaware that approved food additives in food products in Saudi Arabia are considered safe for health. Most participants (54.1%) reported not worrying about their children developing diseases after consuming foods with additives, while 54.7% expressed no fear of their children developing allergies after consuming such foods. Additionally, 52.6% of mothers were willing to pay more money for food without additives. Finally, 90.2% believed that implementing awareness programs in the community regarding food additives is crucial. Overall, most mothers had fair attitudes (56.0%) while a smaller proportion had good (23.8%) attitudes toward food additives (Table 2).

## Mothers' purchasing behavior

Table 3 presents the mothers' responses to questions regarding their purchasing behaviors. The highest percentages were related to health considerations, with 91.7% (the highest) of mothers expressing a preference for natural products for their children. In contrast, the lowest percentage, 57.4%, relates to mothers who reported that their children could not freely choose their food and drinks. These results show that most mothers (78.5%) exhibited good purchasing behavior regarding food additives (7.03 ± 0.732) (Table 3).

**Table 3  Total score of mothers purchasing behavior.**

|       |      | n   | %     | Minimum | Maximum | Mean | Std. Deviation |
|-------|------|-----|-------|---------|---------|------|----------------|
|       | Poor | 20  | 3.8   | 1       | 2       | 1.40 | .503           |
| Level | Fair | 92  | 17.7  | 3       | 5       | 4.33 | .758           |
|       | Good | 409 | 78.5  | 6       | 8       | 7.03 | .732           |
| Total |      | 521 | 100.0 | 1       | 8       | 6.34 | 1.601          |

**Notes.**
The responses for these questions were "Always", "Sometime", "Rarely" and "Never." If respondents answered, "Always or Sometimes", then they were classified as having a positive purchasing behavior, and if the answer was "Rarely or Never", they were classified as having a negative purchasing behavior, depending on the questions. Furthermore, 1 point was given for each correct response and 0 points were given for each incorrect response. The scoring ranges from 0 to 8 points, with points between 8 to 6 points considered as having a good purchasing behavior, points from 5 to 3 points considered having fair purchasing behavior and from 2 to 0 considered having a poor purchasing behavior.

### Dietary patterns of preschool children

Table 4 presents responses to the dietary patterns of preschool children. The overall mean score was $3.95 \pm 1.36$ out of 7-point Likert scale. The highest mean score was $4.98 \pm 1.50$ for biscuits and crackers (including corn and wheat crackers and all kinds of biscuits), with 36.7% of children consuming them once daily. The lowest mean score was $2.73 \pm 2.04$ for soft beverages (*e.g.*, Pepsi, 7UP), with 46.6% of children reporting never consuming them.

### Differences in mothers' knowledge of food additives according to demographic variables

The results indicated that there were statistically significant differences between mothers' knowledge and their age, education level, occupation status, and economic status ($p < 0.05$). Marital status also showed a weak difference ($p = 0.093$) at the 0.10 level. *Post-hoc* multiple comparisons (Scheffé's method) tests reveal that the significant difference between the mothers' ages favored older mothers, indicating that knowledge of food additives increases with age. Regarding education, the significant difference favored those with postgraduate studies, who had the highest mean score of knowledge of food additives ($5.72 \pm 2.083$). Regarding occupation status, employed mothers had the highest mean score of knowledge of food additives ($5.44 \pm 2.371$). Regarding income, mothers with 8,000 SR or more had a mean score greater than 5 out of 8 degrees, higher than for low-income mothers. No significant differences ($p > 0.05$) were found in mothers' knowledge of food additives for the other demographic variables (Table 5).

### Differences in mothers' attitudes toward food additives according to demographic variables

Table 6 presents the ANOVA results for differences in mothers' attitudes toward food additives according to demographic variables. A statistically significant difference was observed between occupation status groups ($p < 0.05$). *Post-hoc* multiple comparisons (Scheffé's method) results show a significant difference between employee mothers and both housewives and students in favor of employee mothers, namely indicating that the higher the mother's career status, the more positive her attitudes toward food additives. No significant difference was found ($p > 0.05$) in mothers' attitudes toward food additives according to the other demographic variables.

**Table 4  Participants' responses towards dietary pattern of preschool children.**

| | | Never (1) | Once or <monthly (2) | Once weekly (3) | 2–3 weekly (4) | 4–5/ weekly (5) | Once/ day (6) | More than once/day (7) | Mean | Std. Deviation |
|---|---|---|---|---|---|---|---|---|---|---|
| Biscuits and crackers (corn and wheat crackers and all kinds of biscuits) | n | 6 | 25 | 68 | 107 | 55 | 191 | 69 | 4.98 | 1.50 |
| | % | 1.2 | 4.8 | 13.1 | 20.5 | 10.6 | 36.7 | 13.2 | | |
| Potato chips (all kinds of chips such as Cheetos, Lays, Doritos, *etc.*) | n | 21 | 45 | 106 | 113 | 60 | 133 | 43 | 4.38 | 1.63 |
| | % | 4.0 | 8.6 | 20.3 | 21.7 | 11.5 | 25.5 | 8.3 | | |
| Cakes (Pancake, cheesecake, chocolate cake) | n | 10 | 46 | 89 | 116 | 74 | 140 | 46 | 4.54 | 1.56 |
| | % | 1.9 | 8.8 | 17.1 | 22.3 | 14.2 | 26.9 | 8.8 | | |
| Breakfast cereals (all kinds of corn flakes) | n | 42 | 51 | 60 | 93 | 65 | 150 | 60 | 4.49 | 1.82 |
| | % | 8.1 | 9.8 | 11.5 | 17.9 | 12.5 | 28.8 | 11.5 | | |
| Fortified dairy products (yogurt with berries, chocolate milk, *etc.*) | n | 42 | 40 | 42 | 67 | 71 | 160 | 99 | 4.84 | 1.87 |
| | % | 8.1 | 7.7 | 8.1 | 12.9 | 13.6 | 30.7 | 19.0 | | |
| Chewing gums (Bubble gum, ball gum, *etc.*) | n | 156 | 65 | 69 | 74 | 42 | 79 | 36 | 3.31 | 2.04 |
| | % | 29.9 | 12.5 | 13.2 | 14.2 | 8.1 | 15.2 | 6.9 | | |
| Juices (canned juices such as Suntop, Nectar, *etc.*) | n | 56 | 45 | 59 | 82 | 67 | 138 | 74 | 4.48 | 1.92 |
| | % | 10.7 | 8.6 | 11.3 | 15.7 | 12.9 | 26.5 | 14.2 | | |
| Soft beverages (Pepsi, 7UP, *etc.*) | n | 243 | 57 | 48 | 63 | 27 | 46 | 37 | 2.73 | 2.04 |
| | % | 46.6 | 10.9 | 9.2 | 12.1 | 5.2 | 8.8 | 7.1 | | |
| Candies (lollipops, *etc.*) | n | 69 | 90 | 88 | 78 | 71 | 83 | 42 | 3.79 | 1.87 |
| | % | 13.2 | 17.3 | 16.9 | 15.0 | 13.6 | 15.9 | 8.1 | | |
| Chocolate (of all kinds) | n | 19 | 53 | 82 | 92 | 89 | 130 | 56 | 4.52 | 1.66 |
| | % | 3.6 | 10.2 | 15.7 | 17.7 | 17.1 | 25.0 | 10.7 | | |
| Ice cream (Quality, Saudia, berries, fruits, chocolate, *etc.*) | n | 40 | 112 | 90 | 94 | 60 | 85 | 40 | 3.84 | 1.78 |
| | % | 7.7 | 21.5 | 17.3 | 18.0 | 11.5 | 16.3 | 7.7 | | |
| Jelly (sweet or sour jelly) | n | 164 | 92 | 54 | 69 | 47 | 60 | 35 | 3.12 | 2.01 |
| | % | 31.5 | 17.7 | 10.4 | 13.2 | 9.0 | 11.5 | 6.7 | | |
| Processed meat (sausage, mortadella, chicken nuggets) | n | 166 | 101 | 58 | 60 | 36 | 69 | 31 | 3.06 | 2.00 |
| | % | 31.9 | 19.4 | 11.1 | 11.5 | 6.9 | 13.2 | 6.0 | | |
| Sauces (mayonnaise, k*etc*hup, ranch dressing, *etc.*) | n | 89 | 85 | 74 | 87 | 56 | 85 | 45 | 3.71 | 1.95 |
| | % | 17.1 | 16.3 | 14.2 | 16.7 | 10.7 | 16.3 | 8.6 | | |
| Instant noodles and soups (Indomie, Maggi, *etc.*) | n | 110 | 99 | 76 | 63 | 55 | 79 | 39 | 3.47 | 1.98 |
| | % | 21.1 | 19.0 | 14.6 | 12.1 | 10.6 | 15.2 | 7.5 | | |
| Overall score for dietary pattern of preschool children | | | | | | | | | 3.95 | 1.36 |

## Relation between food additives related knowledge, attitude, and willingness to buy food additives among mothers in the western region of Saudi Arabia

There was a significant positive relationship between mothers' knowledge of food additives and both attitudes and purchasing behaviors, with correlation coefficients of $r = 0.293$ and $r = 0.284$, respectively (Table 7). This suggests that greater knowledge about food additives was associated with more positive attitudes and purchasing behaviors. Conversely, a significant negative relationship was observed between the dietary patterns of preschool children and both knowledge and attitudes, with correlation coefficients of $r = -0.100$ and

**Table 5  Differences in mothers' knowledge toward food additives according to demographic variables.**

| Demographic variables | Groups | N | Mean | Std. Deviation | F-ANOVA | p-value |
|---|---|---|---|---|---|---|
| Age of mother | 18–22 years old | 39 | 3.50 | 2.769 | 2.181 | 0.043[*] |
|  | 23–27 years old | 89 | 4.90 | 2.506 |  |  |
|  | 28–31 years old | 130 | 4.93 | 2.349 |  |  |
|  | 32–36 years old | 115 | 5.14 | 2.464 |  |  |
|  | 37–41 years old | 77 | 5.13 | 2.489 |  |  |
|  | 42–46 years old | 43 | 4.44 | 2.906 |  |  |
|  | Above 46 years old | 28 | 4.46 | 2.281 |  |  |
| Mother's education level | Less than high school | 30 | 4.07 | 2.504 | 3.002 | 0.018[*] |
|  | High school | 97 | 4.53 | 2.525 |  |  |
|  | Diploma | 38 | 4.84 | 2.563 |  |  |
|  | Bachelor | 295 | 4.86 | 2.559 |  |  |
|  | Postgraduate studies | 61 | 5.72 | 2.083 |  |  |
| Marital status of mother | Married | 490 | 4.87 | 2.510 | 2.386 | 0.093 |
|  | Divorced | 25 | 5.08 | 2.629 |  |  |
|  | Widow | 6 | 2.67 | 1.966 |  |  |
| Mother Occupation status | Businesswoman | 27 | 4.93 | 2.495 | 4.698 | 0.003[**] |
|  | Employee | 160 | 5.44 | 2.371 |  |  |
|  | Housewife | 288 | 4.55 | 2.587 |  |  |
|  | Students | 43 | 4.53 | 2.271 |  |  |
| Economic situations | Under 4,000 SR per month | 52 | 4.23 | 2.587 | 3.178 | 0.014[*] |
|  | 4,000–less than 8,000 SR per month | 173 | 4.47 | 2.587 |  |  |
|  | 8,000 –less than 15,000 SR per month | 202 | 5.20 | 2.398 |  |  |
|  | 15,000–less than 25,000 SR per month | 63 | 5.16 | 2.451 |  |  |
|  | Above 25,000 SR per month | 31 | 5.19 | 2.562 |  |  |
| The city you live (in west region) | Jeddah city | 330 | 4.87 | 2.423 | 0.047 | 0.954 |
|  | Makkah city | 100 | 4.87 | 2.766 |  |  |
|  | Other | 91 | 4.78 | 2.598 |  |  |
| Number of preschool children (from 3 to 5 years) | 1 | 348 | 4.81 | 2.535 | 1.278 | 0.281 |
|  | 2 | 113 | 5.20 | 2.450 |  |  |
|  | 3 | 25 | 4.44 | 2.293 |  |  |
|  | More than 3 | 35 | 4.46 | 2.683 |  |  |
| Child Age | 3 years | 180 | 4.71 | 2.571 | 1.112 | 0.330 |
|  | 4 years | 119 | 4.71 | 2.491 |  |  |
|  | 5 years | 222 | 5.05 | 2.488 |  |  |
| BMI | Underweight range | 272 | 5.01 | 2.440 | 1.024 | 0.382 |
|  | Healthy weight range | 72 | 4.78 | 2.634 |  |  |
|  | Overweight range | 20 | 4.80 | 2.931 |  |  |
|  | Obesity range | 60 | 4.40 | 2.650 |  |  |

**Notes.**
[**] Difference was significant at the 0.01 level.
[*] Difference was significant at the 0.05 level.

**Table 6  Differences in mothers' attitudes towards food additives according to demographic variables.**

| Demographic variables | Groups | N | Mean | Std. Deviation | F-ANOVA | *p*-value |
|---|---|---|---|---|---|---|
| Age of mother | 18–22 years old | 39 | 4.03 | 1.912 | | |
| | 23–27 years old | 89 | 4.02 | 1.777 | | |
| | 28–31 years old | 130 | 4.17 | 1.662 | | |
| | 32–36 years old | 115 | 4.12 | 1.788 | 0.443 | 0.850 |
| | 37–41 years old | 77 | 3.96 | 1.626 | | |
| | 42–46 years old | 43 | 3.93 | 2.028 | | |
| | Above 46 years old | 28 | 3.64 | 2.004 | | |
| Mother's education level | Less than high school | 30 | 3.67 | 1.561 | | |
| | High school | 97 | 3.96 | 1.887 | | |
| | Diploma | 38 | 4.08 | 1.792 | 0.839 | 0.501 |
| | Bachelor | 295 | 4.04 | 1.781 | | |
| | Postgraduate studies | 61 | 4.34 | 1.601 | | |
| Marital status of mother | Married | 490 | 4.02 | 1.776 | | |
| | Divorced | 25 | 4.52 | 1.661 | 1.071 | 0.343 |
| | Widow | 6 | 3.67 | 1.633 | | |
| Mother Occupation status | Businesswoman | 27 | 4.37 | 1.363 | | |
| | Employee | 160 | 4.43 | 1.754 | 5.073 | 0.002[**] |
| | Housewife | 288 | 3.85 | 1.744 | | |
| | Students | 43 | 3.60 | 1.904 | | |
| Economic situations | Under 4,000 SR per month | 52 | 3.87 | 1.794 | | |
| | 4,000–less than 8,000 SR per month | 173 | 4.13 | 1.849 | | |
| | 8,000 –less than 15,000 SR per month | 202 | 4.02 | 1.670 | 0.457 | 0.767 |
| | 15,000–less than 25,000 SR per month | 63 | 3.90 | 1.748 | | |
| | Above 25,000 SR per month | 31 | 4.26 | 1.999 | | |
| The city you live (in west region) | Jeddah city | 330 | 4.08 | 1.796 | | |
| | Makkah city | 100 | 3.97 | 1.795 | 0.156 | 0.856 |
| | Other | 91 | 4.01 | 1.657 | | |
| Number of preschool children (from 3 to 5 years) | 1 | 348 | 4.05 | 1.738 | | |
| | 2 | 113 | 4.01 | 1.835 | 0.056 | 0.983 |
| | 3 | 25 | 3.96 | 1.947 | | |
| | More than 3 | 35 | 4.11 | 1.811 | | |
| Child Age | 3 years | 180 | 4.04 | 1.798 | | |
| | 4 years | 119 | 4.29 | 1.861 | 1.790 | 0.168 |
| | 5 years | 222 | 3.91 | 1.688 | | |
| BMI | Underweight range | 272 | 3.98 | 1.740 | | |
| | Healthy weight range | 72 | 4.21 | 1.768 | 0.330 | 0.803 |
| | Overweight range | 20 | 4.05 | 1.761 | | |
| | Obesity range | 60 | 4.00 | 1.657 | | |

**Notes.**
[**]Difference was significant at the 0.01 level.
[*]Difference was significant at the 0.05 level.

**Table 7 Relation results between food additives related knowledge, attitudes, and willingness to buy food additives among mothers in western region of Saudi Arabia.**

| | | Total score of mothers' knowledge toward food additives | Total score of mothers' attitudes towards food additives | Total score of mothers purchasing behavior | Dietary pattern of preschool children |
|---|---|---|---|---|---|
| Total score of mothers' knowledge toward food additives | Pearson Correlation | 1 | .293** | .284** | −.100* |
| | p-value | | 0.000 | 0.000 | 0.022 |
| | N | | 521 | 521 | 521 |
| Total score of mothers' attitudes towards food additives | Pearson Correlation | | 1 | .321** | −0.073 |
| | p-value | | | 0.000 | 0.097 |
| | N | | | 521 | 521 |
| Total score of mothers purchasing behavior | Pearson Correlation | | | 1 | −.333** |
| | p-value | | | | 0.000 |
| | N | | | | 521 |
| Dietary pattern of preschool children | Pearson Correlation | | | | 1 |
| | p-value | | | | |
| | N | | | | |

**Notes.**
**Correlation was significant at the 0.01 level.
*Correlation was significant at the 0.05 level.

$r = -0.333$, respectively. This suggests that greater knowledge and more positive attitudes toward food additives were associated with a lower likelihood of unhealthy dietary patterns in preschool children.

## DISCUSSION

This study aimed to assess Saudi mothers' knowledge, attitudes, and purchasing behaviors regarding food additives, along with the dietary patterns of preschool children. The findings indicate that mothers exhibited a relatively good level of knowledge, fair attitudes, positive purchasing behaviors, as well as favorable dietary patterns for their preschool children. These outcomes were found to be influenced by various demographic characteristics. Mothers aged between 28 and 31, employed mothers, and those with higher incomes (above 25,000 SR per month) displayed significantly greater knowledge. These results differ from a study conducted in Egypt, where most mothers exhibited poor knowledge about food additives, although educated mothers with higher socio-economic status demonstrated better understanding about food additives (*Hamed, Ferer & Nofal, 2021*).

It is worth noting that younger and more educated mothers were often more knowledgeable about trends in the food industry. They were more attentive to reading the recommendations in the nutrition field and understanding the potential impact of food additives on their child's health. Moreover, educated mothers are skilled at making the right decisions when selecting healthy food choices and can distinguish the ingredients of various food products. Additionally, they might be more proficient in seeking out reliable information and understanding complex nutritional information (*Zugravu, 2012*).

Research has shown that employed mothers with higher financial capabilities may also have the opportunity to attend scientific classes or conferences related to the food industry and food additives, increasing their knowledge and benefitting their preschool-aged children (*Ravikumar et al., 2022*).

The results suggest that occupation status significantly impacts attitudes toward food additives. Employee mothers tend to have good attitudes toward food additives, whereas housewives, businesswomen, and students often exhibit poor or fair attitudes, possessing only basic information about additives without going into details. This could be explained by the fact that working mothers may gain experience and exchange new information with peers in their jobs (*Nsiah-Asamoah, Pereko & Intiful, 2019*). In contrast, a study conducted in Oman found that non-working Omani mothers have a positive attitude toward healthy dietary intake for their children, spending more time at home, caring for their children, and being able to prepare traditional meals, thereby avoiding processed foods (*Al-Shookri et al., 2011*). Conversely, the stress experienced by mothers working full-time jobs may result in a negative attitude toward maintaining a healthy food pattern for their children compared to mothers without jobs or with part-time jobs. Long work hours may prevent mothers from seeking updates and information about food technology for their children's nutrition (*Ergün & Bozdemir, 2023*). Previous multicenter studies in European countries such as Belgium, Cyprus, Estonia, Germany, Hungary, Italy, Spain, and Sweden have shown that a higher educational level of parents is associated with a clearer attitude toward food additives for their children (*Jilani et al., 2018*). Additionally, according to *Hamed, Ferer & Nofal (2021)*, positive attitudes among mothers have been linked to a higher socio-economic status. This could be explained by the fact that employed mothers may have higher purchasing power, which could affect their buying choices. In addition, they might be more willing to purchase products with additives if they perceive these products as offering better quality or if they believe that the additives are safe and beneficial. The difference between studies could be explained by variations in geographical location, dietary habits, perceptions, and social and cultural backgrounds.

Furthermore, some countries have placed a strong emphasis on implementing nutrition intervention programs specifically targeting mothers. This focus has contributed to an increased clear attitude toward child products containing processed additives. According to *Setia et al. (2020)*, there was an increase in positive attitudes toward healthy food choices and behaviors among mothers after a two-month nutrition intervention program. A clear attitude among mothers about nutritional information impacts all family members, not just the child, and can lead to changes in eating behaviors toward a healthier lifestyle.

The findings of this study indicate that the majority of mothers exhibited positive purchasing behavior by avoiding food additive products for their preschool children, remaining unaffected by their children's unhealthy requests. These findings align with a previous study conducted in Italy, where mothers refrained from purchasing products containing artificial colors and sweeteners for their children (*Baldassarre, Campo & Falcone, 2016*). However, contrasting findings regarding mothers' purchasing behavior emerged in a study by *Monalisa (2020)*, which indicated that preschool children could influence their mothers' decision-making to purchase products with low nutrient content. This disparity

in results could be explained by variations in mothers' levels of education. Mothers with higher levels of education tend to make healthier decisions when purchasing healthy products and are more aware of the potential side effects of food additive products. Studies have shown that when consumers understand food labeling regulations, such as those related to nutritional information or food safety warnings, they are more likely to use this information to guide their purchasing decisions and select products that align with their health and safety preferences (*Grunert & Wills, 2007*; *Drichoutis, Lazaridis & Nayga, 2006*). Moreover, mothers with strong emotional connections to their children tend to purchase any food products to satisfy their children's desires and ensure their happiness (*Estay, Kurzer & Guinard, 2021*).

Additionally, the attractive packaging of children's products containing food additives not only directly influences children's food preferences but also increases the pressure on mothers during purchasing decisions to satisfy their children's food requests (*Walsh, Meagher-Stewart & Macdonald, 2015*). A study conducted in the United States found that packaging featuring vibrant colors and attractive images significantly influences parents, particularly mothers, in their decision to purchase food additive products for preschool children aged 2 to 5 years, rather than considering taste or health benefits (*Prible, 2017*). Food additive products are also widely popular and readily available in both small and large markets, including urban and rural grocery shops. Furthermore, these products are inexpensive, have a longer shelf life, are often ready to eat, and require minimal preparation time (*Kraemer et al., 2022*).

In the present study, mothers reported the daily frequency of their preschool children's consumption of food additives products, which included biscuits, crackers, potato chips, cakes, breakfast cereals, fortified dairy products, juices, and chocolate. A previous study conducted in Egypt revealed that among preschool children, the products most frequently consumed were fries, biscuits, lollipops, cake, and chocolate (*Hamed, Ferer & Nofal, 2021*). Similarly, a study conducted in the eastern region of Saudi Arabia among preschool children aged between 1 and 5 indicated that they consumed certain food daily, including pizza, burgers, chocolate or dessert, butter-mayonnaise, soft drinks, juices, chips, and biscuits (*Darwish et al., 2014*). Regular consumption of these types of foods not only increases the intake of food additives but also raises the risk of developing food allergies and dental caries due to the high sugar content and potential allergens commonly found in these products. The current findings related to food allergies and dental caries suggest that dietary recommendations should consider these health conditions, guiding parents in making informed choices that address both nutritional needs and health concerns.

Implementing a combination of interventions, policy changes, and educational programs, such as workshops and online resources that provide parents with knowledge about healthy eating and the impact of food additives on health, can empower parents to make better-informed choices for their children. These efforts could improve dietary patterns among preschool children, foster healthier eating habits, and support overall well-being in this age group. Additionally, policies that restrict the marketing of unhealthy foods to young children could reduce their exposure to high-sugar and high-fat products. Furthermore, enhancing food labeling requirements to provide clear and comprehensive

information about additives, sugar content, and nutritional value can further assist parents in making more informed food choices for their children.

To our knowledge, this study is among the first few studies in the Middle East aimed at assessing the knowledge, attitudes, and purchasing behaviors of Saudi mothers regarding food additives, as well as the dietary patterns of preschool children. However, the study has certain limitations. First, being an observational study, it cannot establish causality. Additionally, the use of a self-reported questionnaire introduces the potential for response bias. Furthermore, relying on convenience sampling as a method of data collection limits the generalizability of the findings to the general population of Saudi Arabia.

## CONCLUSIONS

The findings revealed that mothers in the western region of Saudi Arabia had a relatively good level of knowledge and fair attitudes regarding food additives. Consequently, they tended to engage in better purchasing behaviors and adopt healthier dietary patterns for their preschool children.

These mothers demonstrated a preference to prioritize fresh, unprocessed foods and made informed choices that minimized the consumption of products containing artificial additives. Their heightened knowledge and understanding of the potential health risks associated with additives enabled them to make conscious decisions concerning their children's food preferences.

The study identified a close association between a mother's awareness and certain demographic factors such as age, occupation, and high income levels. Younger mothers and those with higher levels of education had greater access to reliable information sources, including healthcare professionals and credible sources, making them more aware of both the presence and potential effects of food additives. This highlights the importance of implementing educational interventions and providing accessible information to empower mothers to make informed dietary choices for their children. Furthermore, a higher income level was observed to contribute to the adoption of healthy dietary practices. Despite the higher costs, these mothers were inclined to purchase products with high nutritional value, reflecting the importance they placed on their children's health.

Collaboration among researchers, policymakers, and food industry stakeholders is vital. Such collaboration could raise public awareness across different segments of society by developing evidence-based food labeling guidelines, implementing nutritional intervention, reducing the use of harmful additives, and promoting the availability of nutritious and additive-free products. These efforts could significantly enhance parents' awareness, attitudes, and purchasing behaviors regarding food additives. Future longitudinal studies are recommended to track changes in knowledge, attitudes, and dietary behaviors related to food additives. This approach could help establish causal relationships and provide a better understanding of the long-term effects of dietary patterns on health outcomes.

## ACKNOWLEDGEMENTS

The authors would like to thank all participants for their involvement.

### Funding

The APC was funded by the Deanship of Scientific Research (DSR) at King Abdulaziz University, Jeddah, under grant no. (GPIP:1226-253-2024). The funders had no role in study design, data collection and analysis, decision to publish, or preparation of the manuscript.

### Grant Disclosures

The following grant information was disclosed by the authors:
the Deanship of Scientific Research (DSR) at King Abdulaziz University, Jeddah: GPIP:1226-253-2024.

### Competing Interests

The authors declare there are no competing interests.

### Author Contributions

- Reem H. Almoabadi conceived and designed the experiments, performed the experiments, analyzed the data, prepared figures and/or tables, authored or reviewed drafts of the article, and approved the final draft.
- Mahitab A. Hanbazaza conceived and designed the experiments, authored or reviewed drafts of the article, and approved the final draft.

### Human Ethics

The following information was supplied relating to ethical approvals (*i.e.*, approving body and any reference numbers):

The research protocol was approved by the Research Ethics Committee, Faculty of Medicine, King Abdulaziz university with reference number (184-22).

### Data Availability

The raw measurements are available in the Supplementary File.

### Supplemental Information

Supplemental information for this article can be found online at http://dx.doi.org/10.7717/peerj.18223#supplemental-information.

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
