# Peer review of "Knowledge, attitude and purchasing behavior of Saudi mothers towards food additives and dietary pattern of preschool children"

_PeerJ, doi:10.7717/peerj.18223_

## Round 0.1 · original submission · Major Revisions

Thank you authors for your patience

Please find the comments raised and attend to them. Remember to provide detailed responses in your rebuttal letter and be sure to incorporate any relevant responses into the revised manuscript

Thank you

·

Basic reporting

I have reviewed the manuscript titled " Knowledge, attitude and purchasing behavior of
Saudi mothers towards food additives and dietary pattern of preschool children." The study was interesting; however, certain areas require improvement, my detailed comments and suggestions are provided below.
For authors I should recommend that’s, please enhance the introduction by providing more recent references to underline the significance of the study.
Clearly state the research gap that this study aims to fill.
For review litrature purpose, expand on the adverse effects of specific food additives with more detailed examples from recent studies.
You mentioned that the study aims to evaluate the knowledge, attitude, and purchasing behavior toward food additives among mothers in the western region of Saudi Arabia. Why was this specific region chosen for the study? Are there particular characteristics or trends in this region that justify its selection?
The introduction lists several food additives and mentions over 10,000 substances classified as food additives. Does your study focus on specific food additives or categories of additives? If so, which ones and why were they chosen?
The introduction suggests that educating mothers about healthy food choices can positively influence their children's diets. Can you elaborate on the specific educational interventions or strategies that have been shown to be effective in previous studies? How does your study aim to contribute to this body of knowledge?
Is there evidence to suggest that regulatory awareness impacts purchasing decisions?
Ensure all references are formatted according to PeerJ guidelines.
Add more recent references to support the introduction and discussion sections.

Experimental design

The study uses a cross-sectional design with an online survey, which is appropriate for the research question. However, I have some concerns as detailed below,
You utilized a nonprobability convenience sample of 385 mothers. Could you provide more detailed justification for the sample size and explain why a convenience sample was chosen over other sampling methods? How might this sampling method impact the generalizability of your findings?
Mention any potential biases introduced by this sampling method and how they were addressed?
The Food Frequency Questionnaire (FFQ) was adopted from a previous study. Could you provide more details on the process of adapting the FFQ to your study? Were there any modifications made to tailor it to the specific context of your research?

Validity of the findings

You excluded 51 responses from participants living outside the western region of Saudi Arabia. Can you provide more details on how the exclusion criteria were determined? Were there any notable differences in the excluded responses that could impact the study's findings?
While 52.1% of participants acquired information about food additives from social media/the internet, can you assess the accuracy and reliability of these sources? How do you think the source of information might affect the participants' knowledge and perceptions of food additives?
The results indicate that a significant percentage of children were underweight or obese, and many suffered from food allergies or dental caries. Can you discuss how these health indicators were measured and their potential implications for the study's findings on dietary patterns?
You found statistically significant differences in mothers' knowledge of food additives based on age, education level, occupation status, and economic status. Can you provide more insight into why these particular demographic factors influence knowledge levels? Are there any interventions or educational programs that could target these specific groups to improve knowledge?
There were 15 missing data points for weight and 94 for height. How did you handle these missing data in your analysis? Did you employ imputation methods, or were these cases excluded from specific analyses?
The results show that employed mothers had more positive attitudes towards food additives. Can you explore possible reasons for this finding? How might employment status influence mothers' attitudes and purchasing behaviour towards food products containing additives?

Comparative analysis with previous studies in the discussion section
The discussion mentions that higher socioeconomic status correlates with better knowledge and attitudes towards food additives. Could you provide more details on how socioeconomic factors specifically contribute to these differences? For example, does income level affect access to healthier food options or educational materials?
The discussion mentions that preschool children consume certain unhealthy food products daily. What specific interventions or policy changes would you recommend to improve dietary patterns among this age group? Could educational programs targeting both mothers and children be effective?
You acknowledge several limitations, including the observational nature of the study and response bias. Can you propose specific methodologies for future research that might address these limitations? For example, would longitudinal studies or randomized controlled trials provide more robust data?

Additional comments

In your ethical considerations section, you mention obtaining ethical approval and informed consent. Can you include the specific protocol reference number for the ethical approval? Additionally, were there any specific ethical challenges you faced during the study, and how were they addressed?

·

Basic reporting

As a reviewer of the manuscript, I must point out that the survey results lack sufficient detail. While I feel that this topic is fundamental and does not provide any novel insights, the study does hold value as health statistics data collected in Saudi Arabia. The conclusions were appropriately stated. However, each table lacks footnotes. Specifically, the scores need to be explained in the footnotes. Readers want to understand the purpose of this paper just by reading the tables, so explanatory footnotes should be included.

Experimental design

Good.

Validity of the findings

Acceptable.

---

## Round 0.2 · accepted · Accept

I agree with reviewers that the revised manuscript is acceptable for publication. Thank you, authors, for your very fine contribution, and for finding PeerJ as your journal of choice. Look forward to your future scholarly contributions. Congratulations

·

Basic reporting

Dear Authors,

Thank you for your diligence in addressing all the suggested changes in your manuscript, "Knowledge, attitude and purchasing behaviour of Saudi mothers towards food additives and dietary pattern of preschool children." I appreciate the effort you put into refining the content, and it is evident that each recommendation has been carefully considered and implemented.
The revisions have strengthened the study, making it even more impactful and aligned with the high standards expected. We are confident that your work will continue to be a valuable contribution to the field.

Experimental design

No further comments.

Validity of the findings

All the suggestions have been rectified successfully.

·

Basic reporting

No comment.

Experimental design

No comment.

Validity of the findings

No comment.

Additional comments

Sufficient corrections were made with respect to the points raised by reviewer 2. This cross-sectional study makes a valuable contribution to maternal and child health in Saudi Arabia. The authors have clearly addressed the concerns raised, enhancing the clarity and quality of the paper.
Additionally, I would like to commend the authors for their dedication to improving the manuscript. Your work on this important topic will undoubtedly contribute to the field, and I encourage you to continue with your basic research.